# Novel Zebrafish Patient-Derived Tumor Xenograft Methodology for Evaluating Efficacy of Immune-Stimulating BCG Therapy in Urinary Bladder Cancer

**DOI:** 10.3390/cells12030508

**Published:** 2023-02-03

**Authors:** Saskia Kowald, Ylva Huge, Decky Tandiono, Zaheer Ali, Gabriela Vazquez-Rodriguez, Anna Erkstam, Anna Fahlgren, Amir Sherif, Yihai Cao, Lasse D. Jensen

**Affiliations:** 1Department of Health, Medicine and Care, Division of Diagnostics and Specialist Medicine, Linköping University, SE-58185 Linköping, Sweden; 2Department of Biomedical and Clinical Sciences, Division of Urology, Linköping University, SE-58185 Linköping, Sweden; 3BioReperia AB, SE-58213 Linköping, Sweden; 4Department of Biomedical and Clinical Sciences, Division of Cell Biology, Linköping University, SE-58185 Linköping, Sweden; 5Department of Surgery and Perioperative Sciences, Urology and Andrology, Umeå University, SE-90187 Umeå, Sweden; 6Department of Microbiology, Tumor and Cell Biology, Karolinska Institute, SE-17165 Stockholm, Sweden

**Keywords:** cancer, BCG, immune-oncology, zebrafish, PDX, xenograft, personalized medicine

## Abstract

Background: Bacillus Calmette-Guérin (BCG) immunotherapy is the standard-of-care adjuvant therapy for non-muscle-invasive bladder cancer in patients at considerable risk of disease recurrence. Although its exact mechanism of action is unknown, BCG significantly reduces this risk in responding patients but is mainly associated with toxic side-effects in those facing treatment resistance. Methods that allow the identification of BCG responders are, therefore, urgently needed. Methods: Fluorescently labelled UM-UC-3 cells and dissociated patient tumor samples were used to establish zebrafish tumor xenograft (ZTX) models. Changes in the relative primary tumor size and cell dissemination to the tail were evaluated via fluorescence microscopy at three days post-implantation. The data were compared to the treatment outcomes of the corresponding patients. Toxicity was evaluated based on gross morphological evaluation of the treated zebrafish larvae. Results: BCG-induced toxicity was avoided by removing the water-soluble fraction of the BCG formulation prior to use. BCG treatment via co-injection with the tumor cells resulted in significant and dose-dependent primary tumor size regression. Heat-inactivation of BCG decreased this effect, while intravenous BCG injections were ineffective. ZTX models were successfully established for six of six patients based on TUR-B biopsies. In two of these models, significant tumor regression was observed, which, in both cases, corresponded to the treatment response in the patients. Conclusions: The observed BCG-related anti-tumor effect indicates that ZTX models might predict the BCG response and thereby improve treatment planning. More experiments and clinical studies are needed, however, to elucidate the BCG mechanism and estimate the predictive value.

## 1. Introduction

Cancer is recognized as a highly heterogeneous class of diseases, where specific disease characteristics including the efficacy of available medical or non-medical treatments differ greatly among patients [1]. Therefore, diagnostic methods that may identify disease characteristics associated with sensitivity to available therapies for each patient are becoming an increasingly important aspect of cancer medicine. Methods used for this purpose include genetic characterization of tumor material to find mutations or gene-expression profiles associated with a heightened response rate [2], but such molecular diagnostics suffer from poor accuracy as many patients do not respond as predicted [3]. This is particularly problematic within the growing field of immune-oncology, where up to two-thirds of the patients that test positive using such molecular tests (e.g., PD-L1^high^ tumors) turn out to be non-responders to the corresponding (e.g., anti-PD-1 or anti-PD-L1) therapy [4]. Compared to molecular tests, functional assessment of drug efficacy using, for example, patient-derived 3D-cell models in vitro (i.e., patient-derived organoids, PDO) or in vivo (i.e., patient-derived xenografts, PDX) have much higher accuracy and correctly predict the treatment outcome for ~80–90% of the patients tested [5,6,7]. PDX models in mice are, however, too costly for routine clinical implementation and are not timebound within the clinical decision-making processes existing for most types of cancer [7]. PDO models are faster and cheaper and could potentially be implemented in the clinical routine. Current PDO technologies, however, are poorly suited for evaluating drugs that target complex interactions between malignant and non-malignant cells in the tumor microenvironment such as those used within immune-oncology. Therefore, fast, functional patient-derived tumor model technologies that allow accurate efficacy predictions of immune-oncology drugs are urgently needed to improve clinical treatment planning.

Urinary bladder cancer is the most prevalent urothelial malignancy and the seventh-most prevalent cancer overall [8]. Due to its high mutational frequency resulting in numerous different molecular drivers of the disease, it is a very heterogeneous disease, for which meaningful molecular sub-classifications have not been clinically established [9]. Instead, these cancers are classified based on medical imaging and pathological findings as low- to high-grade non-muscle-invasive bladder cancer (NMIBC) or muscle-invasive bladder cancer (MIBC) [9,10]. The standard-of-care primary adjuvant therapy for NMIBC (T-categories TaG3 and T1) is intravesical Bacillus Calmette-Guerin (BCG) treatment, which is one of the earliest and best-established immune-oncology therapies applied in cancer [11,12]. BCG is an attenuated strain of *Mycobacterium bovis*, which has been developed and used as a vaccine against *Mycobacterium tuberculosis* since 1921 [12]. The anti-tumor effect of BCG vaccination in malignant diseases was first reported in the 1950s when it was observed that BCG-infected mice show a higher resistance to tumor transplantation [13]. Supported by findings demonstrating a strong delayed hypersensitivity reaction to immunogenic antigens in the bladder of guinea pigs [14], Morales et al. reported in 1976 that intravesical BCG instillation lowered both recurrence and progression rates in nine cases of human bladder cancer [15]. This was further confirmed by Lamm et al. in a randomized prospective trial in 1980 [16], leading to the clinical application of BCG, which, until today, has remained the recommended first-line therapy in intermediate and high-risk NMIBC after transurethral tumor resection [11,17]. However, while reducing the risk of tumor recurrence and progression by up to 30% compared to tumor removal alone, recurrence is seen in approximately 30 to 40% of the patients, and up to 15% will progress to invasive and disseminated disease [18,19,20]. Given this considerable risk of disease recurrence after BCG therapy and frequent cases of unexplainable primary treatment failure, it is important but challenging to identify and manage both BCG-sensitive and BCG-resistant patient groups [10,11,17]. Even though several studies were conducted to investigate molecular signatures associated with BCG response, no biomarkers have been found that may predict the short or long-term treatment outcome [18,19,21]. On top of this unmet clinical problem, repeated periods of BCG shortage have plagued urology clinics over the past few years [20,22]. This persistent shortage has caused an increase in the number of immediate radical cystectomies instead of bladder-sparing BCG immunotherapy in high-risk patients [17,20,22]. There is therefore an urgent need for tests that may help to prioritize patients for BCG immunotherapy such that only those not likely to respond to such therapy could be considered for immediate radical cystectomy.

Embryonic zebrafish tumor xenograft models are gaining popularity as an experimental system that combines the benefits of PDO and PDX models by offering cheap and fast readouts while also allowing studies of the tumor microenvironment and tumor invasion or metastasis [23]. The tumor xenograft models can be based on either cancer cell lines or patient-derived xenografts (PDX) to better represent the heterogeneity of patient tumors, which then are microinjected into the zebrafish larvae where they create 3D microtumors that allow the investigation of drug-induced changes in tumor size and dissemination [6,24,25]. The generation of zebrafish patient-derived tumor xenograft (ZTX) models has been used in the past to predict treatment outcomes to commonly used chemotherapies in colorectal, gastric, and hematological cancers [26,27,28,29]. Bladder cancer has only recently been studied in this system [30], but whether these models sufficiently recapitulate the immune interactions required to study the efficacy of immune-oncological therapies, including BCG therapy, remains to be determined. 

Here, we create ZTX models for urinary bladder cancer and establish the conditions by which such systems can be exploited to investigate BCG treatment efficacy. We identify the importance of physical contact between the BCG bacteria and the tumor cells as well as the viability of the bacteria, and we determine the dose–response relationship of this treatment in the ZTX model. Finally, we demonstrate that the ZTX models could be established from six of six NMIBC patients and correctly predicted two of two positive and one of two negative treatment outcomes to BCG treatment in the four patients that had finished the treatment. 

## 2. Materials and Methods

### 2.1. Reagents

The following reagents were used in the study: NaCl (Sigma, #S5886); MgSO_4_ (VWR, Radnor, PA, USA, #0662), CaCl_2_ (Sigma, Saint Louis, MI, USA, #C8106); KCl (VWR, #26764.232); 1-Phenyl-2-Thiourea (PTU, Alfa Aesar, Haverhill, MA, USA, #L06690); Eagle’s minimum essential medium (EMEM, ATCC, Manassas, WV, USA, #30-2003); fetal bovine serum (FBS, VWR, #97068–085); Penicillin-Streptomycin (Pen/Strep, Biowest, Nuaille, France, #L0022-100); dimethyl sulfoxide (DMSO, Sigma, #276855-100); Trypsin/EDTA (Biowest, #MS0158100U); RPMI-1640 culture medium (Biosera, Nuaille, France, #LM-R1637/500); phosphate-buffered saline (PBS, VWR, #E403-500); dimethyl sulfoxide (DMSO, Sigma, #276855-100); Fast-DiI™ oil (ThermoFisher, Waltham, MA, USA, #D3899); PDX Disruptor-mix (#P01-3001, Bioreperia, Linköping, Sweden); Cell Implantation Resuspension Medium (#P03-2001, Bioreperia, Sweden); ethyl 3-aminobenzoate methanesulfonate (Tricaine, Sigma, #E10521-50).

### 2.2. Zebrafish Breeding and Maintenance 

The transgenic Tg(fli1a:EGFP) strain [31] with eGFP-expressing vasculature was maintained in the zebrafish facility at Linkoping University (Linkoping, Sweden) and used for all experiments. After breeding, fertilized zebrafish eggs were collected in Petri dishes and incubated at 28 °C in an E3 embryo medium (pH 7.2) containing 0.29 g of NaCl, 0.082 g of MgSO_4_, 0.048 g of CaCl_2_, and 0.013 g of KCl per liter of purified water. The E3 embryo medium was supplemented with 0.2 mM PTU to inhibit pigmentation of the larvae.

### 2.3. Cell Culture and Fluorescent Labelling

The human muscle-invasive urinary bladder cancer cell line UM-UC-3 (ATCC, #CRL-1749) was cultured in T75-cell culture flasks in EMEM supplemented with 10% FBS and 1% Pen/Strep under standard cell culture conditions at 37 °C in a humidified incubator containing 5% CO_2_. The cell culture medium was changed every two to three days. Subculturing was performed following incubation of the cells in 1×Trypsin/EDTA at 37 °C for approximately 5 min, the addition of the cell culture medium, and centrifugation at 250× *g* for 5 min. At 70 to 80% confluency, tumor cells were fluorescently labelled using 1,1-dioctadecyl-3,3,3′,3′-tetramethylindocarbocyanine perchlorate (DiI). After washing once with PBS, 10 mL of pre-warmed PBS containing 2 µg/mL Fast-DiI™ dye was added. Following 30 min of incubation at 37 °C, cells detached from the culture flask and the suspension was centrifuged at 250× *g* for 5 min. The cell pellet was washed twice with 5 mL of PBS, and centrifugation was repeated. Labelled cells were resuspended in 1 mL of culture medium and filtered through a 40 µm cell strainer. Cell counts and cell viability were evaluated using trypan blue staining. 

### 2.4. Clinical Tumor Tissue Samples

After written informed consent, tumor tissue from patients with presumed urinary bladder cancer was collected during TUR-B and placed in cryotubes containing 1 mL cryomedium. The tubes were placed in a CoolCell (Corning, Corning, NY, USA) cryopreservation box within 2 h and kept at −80 °C. The patients were pseudonymized by giving the incoming material of every donor patient a unique code, which was subsequently used to identify the corresponding ZTX model. Ethical approval for studies based on primary tumor biopsies was granted to A. Sherif by the Swedish Ethical Review Authority. Information on each patient’s diagnosis, pathological findings, surgical, and BCG treatment outcome and was collected from their medical journals in accordance with the ethical approval. 

Cryopreserved biopsies from six bladder cancer patients known to be treated with BCG were thawed and washed twice in 10 mL of RPMI-1640 medium supplemented with 10% FBS by gentle mixing. Using ethanol-disinfected surgical scissors, the tumor samples then were minced into small 1–2 mm^3^ pieces and 5 mL of the tissue dissociation enzyme mix was added. The tissue piece suspension was transferred to a gentleMACS™ C-tube (Miltenyi Biotec, Bergish Gladbach, Germany, #130096334) and dissociated for 30–60 min on a gentleMACS™ Octo Dissociator set to 37 °C. The homogenous cell suspension was then filtered, and the C-tube and filters were washed with 5 mL of RPMI 1640 supplemented with 2% FBS (RPMI/2% FBS) before centrifuging the sample for 5 min at 300× *g*. The pellet was resuspended in 10 mL RPMI/2% FBS, re-filtered, and re-pelleted. For fluorescent labelling, cells were incubated at 37 °C in 5 mL of RPMI/2% FBS mixed with Fast-DiI™ dye (8 µg/mL). Following pelleting and washing with PBS, the cells were resuspended in 1 mL RPMI/2% FBS. The total number of cells and cell viability were calculated. 

### 2.5. BCG Reconstitution, Toxicity Testing and Treatment in the Xenograft Model

For BCG immunotherapy, one ampule of BCG-medac powder for intravesical suspension (MTnr: 17493, Lot#210332C) was used. According to information from the supplier, the ampule contained 2 × 10^8^ to 3 × 10^9^ freeze-dried viable units of the BCG-RIVM strain, polygeline, glucose anhydrous, and polysorbate 80. All experiments were performed using one and the same ampule thereby ensuring consistency of the bacterial concentration and other parameters that might be different in different ampules. The ampule was stored according to the manufacturer’s instructions and re-sealed after opening to not jeopardize the stability of the bacteria. The BCG dose in the ampule was assumed to be 1.6 × 10^9^ viable units, i.e., the mean of the indicated interval, to allow further calculations of doses in the ZTX models. An appropriate amount of the powder was reconstituted in 50 mL of 0.9% NaCl, centrifuged at 4600× *g* for 20 min, and resuspended in PBS. In some experiments, BCG was used directly after reconstitution without prior centrifugation, as indicated in the text. To establish a dosage–survival curve of BCG in zebrafish larvae, increasing BCG concentrations ranging from 1.88 × 10^6^ to 2.4 × 10^8^ viable units/mL were injected into 48 h old embryos in groups of 15 embryos. Successfully injected embryos were incubated in Petri dishes containing the E3 medium supplemented with PTU at 35.5 °C, a temperature at which *Mycobacterium bovis* bacteria are still viable [32] and able to induce immune activation [33]. Three days post injection (dpi), signs of toxicity, e.g., edema, and survival of the embryos in the different treatment groups were documented. BCG treatment of the zebrafish tumor xenografts was performed via the co-injection of tumor cells and BCG and intravenous injections of BCG. For this, the desired number of viable units was weighed and reconstituted in 0.9% NaCl. The BCG suspension was then centrifuged at 4600× *g* for 20 min and the pelleted bacteria were resuspended in PBS. Labelled tumor cells were resuspended in the BCG solution immediately before injections. Heat inactivation of BCG was conducted according to a previously validated protocol, shown to effectively kill the vast majority of the BCG bacteria [34]. Briefly, the reconstituted BCG bacteria were incubated on a thermo-shaker set to 80 °C for 20 min before co-injection with UM-UC-3 cells. 

### 2.6. Microinjections into 48 h Old Zebrafish Larvae

Fluorescently labelled cells (with or without BCG) were subcutaneously injected into the PVS of 48 h old zebrafish embryos using a micromanipulator. For this, embryos were mechanically dechorionated using micro-surgical forceps. Stained cell suspensions were centrifuged for 5 min at 250× *g* and resuspended in PBS to reach a cell concentration of ~300 cells/nl. Borosilicate glass capillaries (World-Precision instruments, Sarasota, FL, USA, #TW100-4) were pulled using a Narishige (Setagaya City, Tokyo, Japan) PC-10 micropipette puller and filled with 3–5 µL of cell suspension. At the needle tip, a needle opening was created using fine forceps and the droplet size was calibrated. Dechorionated embryos were put on pre-warmed 2% agarose plates and anesthetized with 1 mg/mL Tricaine before microinjecting the tumor cells. Embryos were transferred back to the E3 medium supplemented with PTU following injection. 

Using fluorescence microscopy, successfully injected embryos, which (for PVS injections) showed tumor cells in the PVS but not in the yolk or in the bloodstream, were selected. The selected zebrafish tumor xenografts were incubated at 35.5 °C in 24-well plates containing the E3 medium supplemented with PTU until 3 dpi. 

### 2.7. Analysis of Primary Tumor Size and Cell Dissemination

Fluorescent images of the zebrafish xenograft models were acquired using a Nikon (Konan, Tokyo, Japan) SMZ1500 fluorescence stereomicroscope. Images were taken at 0 dpi and 3 dpi at 30- and 80-fold magnification. Analysis of the images was conducted using the thresholding function of the open-source ImageJ software [35]. Relative tumor size in percent was determined by calculating the ratio of tumor sizes 3 dpi and 0 dpi and multiplying it by 100 for each individual fish. Distant metastases were evaluated by counting the number of cells in the caudal hematopoietic plexus at 3 dpi. 

### 2.8. Statistical Analysis

Graphical data visualization and statistical analysis were conducted using Graphpad Prism 8.0.2 (Graphpad Software Inc., San Diego, CA, USA). Outliers were identified and removed using the ROUT outlier test (Q = 1), and the normality of data was verified by applying the D’Agostino Pearson omnibus normality test. Two group comparisons were conducted using a two-tailed, unpaired Student’s t-test, and three or more groups were compared by applying a multiple comparisons one-way ANOVA. In the case of unequal variances, the respective corrections were applied. The significance threshold α was set to 0.05. Data are shown as mean ± standard deviation, and *n*-values represent the number of zebrafish larvae analyzed per group.

## 3. Results

### 3.1. A Component of the BCG-Medac Drug Formulation Is Toxic to Zebrafish Larvae but Can Be Removed by Centrifugation

As BCG therapy has not been tested in zebrafish larvae in the past, we first investigated the tolerability and toxicity of the clinical BCG formulation in tumor-free zebrafish larvae. Intravenous injection of a BCG dose similar to that used in human intravesical immunotherapy (6 × 10^7^ viable units/mL) in 2-day-old zebrafish larvae revealed a reduced survival rate at 3 dpi and clear signs of toxicity (e.g., edema) in the surviving larvae (Figure 1A,B). To find the maximum tolerated dose for zebrafish larvae, we injected larvae at 2 dpf with a range of concentrations starting at 1.88 × 10^6^ and reaching 2.4 × 10^8^ viable units/mL and monitored survival until 3 dpi. Under these conditions, the LD50 of BCG-medac was 9.5 × 10^7^ viable units/mL, but most embryos survived at doses between 1.88 × 10^6^ and 9.5 × 10^7^ viable units/mL (Figure 1C). Doses beyond 1.5 × 10^7^ viable units/mL led to similar mortality and toxicity profiles as seen using the clinically relevant dose of 6 × 10^7^ viable units/mL (Figure 1A). As the toxic phenotypes were also seen at low doses of the BCG-medac formulation and were not consistent with phenotypes expected from infection with tuberculosis bacteria in zebrafish larvae [36], we hypothesized that an adjuvant component of the formulation rather than the bacteria themselves might be responsible for the observed toxicity. To test this hypothesis, we centrifuged the reconstituted BCG-medac powder and resuspended the pellet containing BCG bacteria in PBS. We then repeated the toxicity dose–response experiment using the supernatant, containing the water-soluble fraction of the formulation, and the “cleaned” BCG fraction. While the injection of the bacteria-free supernatant led to a similar toxicity and mortality profile as the complete reconstituted formulation, injection of the cleaned bacteria alone was fully tolerated and led to no increase in mortality or signs of excessive toxicity even at the highest doses tested (Figure 1D–F). Taken together, these results indicate that the observed toxicity of the complete reconstituted formulation was due to a water-soluble ingredient of the BCG-medac powder and not the BCG bacteria themselves. We therefore decided to continue further experiments using the cleaned bacterial component. 

### 3.2. BCG Leads to Significant Regression of UM-UC-3 Xenografts

To investigate the effect of BCG on bladder cancer xenografts in the ZTX model, four different concentrations of BCG ranging from 0.5× and up to 2× the concentration used in human intravesical therapy were co-injected with ~300 fluorescently labelled UM-UC-3 [37] cells/nl into the PVS of 48 h old embryos. The relative proportions of tumor cells to BCG bacteria for these experiments are indicated in Appendix A. Relative tumor regression compared to untreated control fish was documented at 3 dpi (Figure 2A). All four doses induced a significant regression in the xenograft sizes (ANOVA: *p* < 0.0001, Figure 2B,C), showing a dose-dependent increase in efficacy from a mean relative tumor size of 50.5% at 3 × 10^7^ to 23.2% at 1.2 × 10^8^ viable units/mL. 

### 3.3. Bacterial Viability and Physical Contact with the Tumor Cells Are Both Required for Efficacy 

We next investigated the mechanism leading to a therapeutic outcome from BCG treatment by testing whether heat-inactivated bacteria, which would still be able to activate part of, but likely not the complete, immune response [38], could provide a similar outcome as the viable bacteria used clinically. While the heat-inactivated BCG treatment significantly increased tumor regression compared to non-treated controls, this effect was greatly attenuated as compared to treatment with the viable BCG therapy (78% vs. 25%, respectively) (Figure 3A–C). This suggests that bacterial viability is important for mounting the complete anti-tumor response in the ZTX models. 

Next, we asked if intravenous administration of viable BCG bacteria, likely leading to systemic rather than tumor-localized immune activation, could also be effective. To investigate this important question, we first established subcutaneous UM-UC-3 xenografts and subsequently injected the BCG therapy intravenously. Interestingly, intravenous injection of BCG was completely ineffective and unable to induce regression of the tumors compared to non-treated controls (Figure 3D,E). Taken together, these findings clearly suggest that complete and localized activation of anti-cancer immunity in the tumor microenvironment is required for optimal efficacy of the BCG therapy. 

### 3.4. BCG Therapy Does Not Impact on Bladder Cancer Invasiveness and Dissemination

Activation of the innate and adaptive immune system has been suggested to impact tumor cell dissemination and metastasis in a complex manner, either inducing or inhibiting metastasis depending on poorly understood, context-specific cues [39,40,41]. We therefore investigated if the complex immune activation caused by either heat-inactivated or viable BCG treatments and elicited either specifically in the tumor microenvironment or systemically in the zebrafish larvae could impact the dissemination of the UM-UC-3 cells (Figure 4A). Dissemination was investigated at the major metastatic site in the caudal hematopoietic plexus, at the caudoventral region of the body (Figure 4A). UM-UC-3 cells readily disseminated to this region under control conditions, and neither heat-inactivated nor viable BCG treatment had any impact on this dissemination phenotype (Figure 4B,C). Furthermore, i.v. injection of the BCG, which is expected to primarily activate the immune system at the caudal hematopoietic plexus where the majority of immune cells reside at these developmental stages [42], did not affect the dissemination and homing of tumor cells to this area (Figure 4B,C). These findings suggest that the type of immunity activated by BCG treatment within the ZTX system is neither pro- nor anti-metastatic. 

### 3.5. ZTX Models Predicted Responses to BCG Therapy in NMIBC Patients 

Next, we asked if the heterogeneity in responses to BCG treatment among bladder cancer patients could be recapitulated in the ZTX platform. To answer this, we obtained cryopreserved, viable tumor tissue samples from six non-muscle invasive bladder cancer patients, known to have been treated with BCG and in four of which the treatment had been completed and the clinical treatment outcome was available. A summary of relevant clinical data related to these six patients is shown in Table 1. The tumor tissues were dissociated, fluorescently labelled, and injected into the PVS of 48 h old zebrafish larvae with or without 6 × 10^7^ viable units/mL BCG (Figure 5A). All six tumors implanted effectively in the zebrafish larvae, exhibiting only minimal spontaneous regression to a mean relative tumor size of 80.8%, 57.1%, 77.1%, 51.6%, 31.1%, and 40.0%, respectively, in the non-treated control groups at 3 dpi (Figure 5B). BCG treatment led to significant additional regression compared to the control group in two of the six ZTX models (Figure 5D,E). For Patient B, the mean relative regression compared to the control was 82%, whereas the mean regression was 32% for Patient C. Both of these patients had responded to BCG treatment; Patient B only had small in situ lesions left at follow-up (which, in clinical practice, is regarded as an objective response), whereas Patient C was a complete responder. The ZTX models generated from Patients A and D, however, did not demonstrate significant regression in the BCG-treatment group, suggesting that these tumors would not respond to BCG treatment (Figure 5C,F). While Patient D was considered a clinical responder, Patient A, indeed, developed recurrent disease associated with clinical BCG treatment failure, confirming the ZTX model results for this patient. ZTX models generated from Patients E and F suggested that BCG treatment would not be effective for these patients as the relative tumor sizes of the treatment groups were significantly larger compared to the controls (Figure 5G,H). While Patient E has started the induction treatment and the outcome of that is still pending, Patient F had recurrence and re-resection of the tumor prior to starting the BCG treatment, which is therefore still pending. Taken together, these findings provide strong, preliminary evidence and proof-of-concept data for predicting BCG treatment outcomes in NMIBC patients using ZTX models. 

## 4. Discussion

Functional experimental model systems such as 3D spheroid/organoid and xenograft models are considered the most accurate and reliable readouts of anti-proliferative, cell killing, or anti-migratory effects of anti-cancer compounds [3]. Because of this, such models have long been the gold standard for anti-cancer efficacy studies in cancer research and pre-clinical drug development [6]. Due to the slow growth of many patient tumor cells in vitro or after implantation in immunocompromised mice, 3D culture or xenograft models based on primary tumor cells take weeks or even months to generate results [7] and have therefore not been successfully implemented for precision diagnostics and individualized treatment planning in cancer patients. Furthermore, current PDO or PDX models do not readily recapitulate the complexity of the immune–tumor cell interactions required to evaluate the efficacy of immune-oncological treatments [43]. While human-like immunity can be established in immune-compromised mice to allow evaluation of some immune-oncology drugs such as checkpoint inhibitors, these models generally retain mouse innate immunity, which may interfere with the readout. Similarly, PDO models that contain T-cells or other immune cells to mirror part of the intratumoral immune system can be generated, but these generally do not allow studies on immune-cell homing to the tumor, tumor cell dissemination, and metastasis or off-tumor targets (such as targets in the lymph nodes), which have high clinical importance. Furthermore, simpler models analyzing tumor cells growing in 2D in vitro do not allow the evaluation of important but complex immune-oncological mechanisms, nor are they well suited for the evaluation of metastatic dissemination through the circulation to distal tissues. 

Zebrafish xenograft models have recently been developed as an alternative to both 2D cell cultures and PDO/PDX models. These models are often constructed to include either specific aspects of human tumor–immune cell interactions such as macrophage- or neutrophil-induced metastasis [39,40,44], T-cell mediated cytotoxicity [45], or macrophage-induced resistance to immune-oncology treatments [46]; alternatively, they can be created to include the entire complexity of the cellular tumor microenvironment [24]. Here we take the zebrafish xenograft platform one step further by demonstrating that the efficacy of a classical immune-oncology therapy for bladder cancer, BCG, can be studied using both established cell lines (CDX models) and patient material (PDX models). While only six patient samples have been studied in this work, of which three were responders, one was a non-responder and two for whom the clinical outcome is still unknown, the ZTX models could be established from all six and correctly predicted the patient treatment outcome in three of four cases, providing a robust proof-of-principle for these models as a system that recapitulates the complex tumor–immune cell interactions of the patients needed to elicit efficacy of the BCG treatment (or the lack thereof). Since we found that direct contact between tumor cells and BCG bacteria was required for efficacy, it is tempting to speculate that the ability of the tumor cells to ingest the bacteria might be involved in mediating a positive treatment outcome. As BCG ingestion depends on the specific oncogenic signaling pathways of the tumor cells [47], this process could have been impaired in the non-responding patient sample. Further mechanistic studies using a larger patient cohort are needed to further address this important clinical question. 

To establish the zebrafish xenograft platform for studies within immune-oncology in general, and for evaluating BCG treatment efficacy in particular, we thoroughly characterized the methodological requirements of the assay. Firstly, we found that co-injection of the clinical BCG-medac formulation led to high mortality of the injected zebrafish embryos. The BCG-medac powder is formulated to include anhydrous glucose, polygeline, and polysorbate 80, in addition to the BCG bacteria themselves. Polygeline, a urea polymer derived via the degradation of gelatin, is a commonly used drug carrier with properties similar to albumin and is held to be unproblematic due to its quick metabolism [48]. Furthermore, glucose has been evaluated in the zebrafish system and found to not cause any toxicity at the concentrations relevant to this study [49]. Polysorbate 80 is a non-ionic surfactant, which is applied as a solubilizing agent. Although it is commonly used in human drug formulations, polysorbate-80-containing injectables have been reported to occasionally cause severe anaphylactoid reactions and hemolysis [50]. Intraperitoneal injections of polysorbate 80 into zebrafish embryos similarly revealed anaphylactoid reactions and increased mortality exceeding 40% of the larvae [51]. In addition to polysorbate-80, the temperature used in this study (35.5 °C) may by itself cause transient toxic phenotypes including pericardial edema, but only if the embryos are exposed to such temperatures from early embryonic stages. Toxicity to this temperature is not seen if exposure is delayed until the larval stages as seen here. On the other hand, the temperature is not likely to have mediated the lack of toxicity of the pelleted and reconstituted bacteria, as the same temperature was used for both formulations. This strongly suggests that polysorbate-80 likely caused the observed toxicity. However, by removing this compound by pelleting the bacteria, the drug formulation could be used without overt toxicity.

Prior in vitro studies have demonstrated a direct BCG-related inhibition of proliferation or BCG-related cell death in certain bladder cancer cell lines but only a few in vivo studies exist [11]. In the zebrafish xenograft model of UM-UC-3, we found that BCG treatment augmented tumor regression to a maximum of 23.2% for the highest BCG concentration. UM-UC-3 is a low-grade, muscle-invasive bladder cancer cell line, which is known to readily internalize BCG by macropinocytosis [47]. BCG immune cell priming alone does not enhance NK or T cell cytotoxicity against UM-UC-3 cancer cells in vitro [52]. Since the mechanism behind the effect of BCG in the ZTX models is completely unknown, some crucial requirements were tested that are known to be important for the efficacy of human BCG therapy. One of these is the direct contact between the BCG bacteria and the urothelium/bladder cancer cells [53]. As intravenous injection of BCG instead of co-injection failed to reduce the relative tumor size of the xenografts, direct BCG–cell contact was shown to also be important in the ZTX models. The second requirement pertains to using therapy containing viable BCG bacteria [11,54]. In this study, heat inactivation of BCG before co-injection significantly reduced the efficacy of the BCG therapy compared to viable BCG. Our heat-inactivation protocol has previously been shown to inactivate the vast majority of the *Mycobacterium bovis* BCG bacilli [34,55]. However, even though the effect was small, inactivated BCG was able to cause a slight tumor regression in the zebrafish CDX model. A probable explanation for this might be that heat-inactivated bacteria still induce a partial immune response, which in turn could eliminate some of the tumor cells. In this context, studies have also shown that both heat-inactivated BCG and sub-cellular BCG fractions stimulate peripheral blood mononuclear cells and induce a cytotoxic NK cell response, which recognizes and eliminates bladder cancer cells [55]. 

A major risk factor for the progression of NMIBC to more advanced stages is local or distal invasion/metastasis of the tumor cells. Here we found that both local and intravenous administration of viable or heat-inactivated BCG did not reduce the invasiveness and initial metastatic dissemination of the tumor cells in the zebrafish larvae. Previously, tumor dissemination in the ZTX models has been shown to accurately predict positive versus negative lymph node disease in non-small cell lung cancer [24], and radiation-induced invasiveness [56] or hypoxia-induced metastasis [25] of various cancers. The mechanisms underlying phenotypic switching of NMIBC to more advanced stages, as well as how this could be affected by BCG treatment, are currently not known. It is theoretically possible that the lack of an effect of BCG on this process in the ZTX models is due to the metastasized cells escaping the primary tumor environment quickly after implantation, prior to internalizing the bacteria. This might prevent immune-mediated tumor cell killing at the metastatic niche. It is also possible that mechanisms that, in a subset of the tumor cells, inhibit the internalization of the bacteria may also be involved in metastasis, implying that the BCG-resistant tumor cells are those that are also metastasizing. Alternatively, mechanisms involved in endowing tumor cells with invasive phenotypes could also lead to immune evasion and, as such, allow metastasized tumor cells with internalized bacteria to stay alive at the metastatic site even if the primary tumor is effectively regressed. These critical questions related to the progression of non-muscle invasive bladder cancer should be the subject of further study using the ZTX models in the future.

## 5. Conclusions

We have demonstrated that BCG co-injection in zebrafish xenograft models of urinary bladder cancer leads to significant tumor regression in both the UM-UC-3 cell line and two out of six clinical patient samples, in both cases recapitulating the clinical treatment outcome of the patients. This indicates that ZTX models might be suitable for predicting the BCG outcome and thereby for BCG treatment planning. However, further investigations on how this mechanism is mediated in the zebrafish model, as well as further clinical validation studies, are needed. 

## Figures and Tables

**Figure 1 cells-12-00508-f001:**
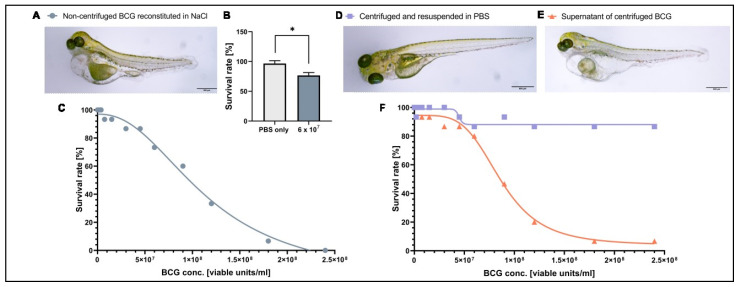
Toxicity of BCG-Medac is caused by a soluble component of the powder. (**A**) Light micrograph image of 5-day-old zebrafish larvae three days after having received a subcutaneous injection of ~3 nL of completely reconstituted BCG-Medac. (**B**) Quantification of larval survival at 5 days post-fertilization, three days after subcutaneous injection of ~3 nL of completely reconstituted BCG-Medac, or only PBS (vehicle). (**C**) Graph showing the proportion of larvae that survived at 5 days post-fertilization, three days after injection with the indicated concentrations of 3 nL of completely reconstituted BCG-Medac. (**D**,**E**) Light micrograph images of 5-day-old zebrafish larvae three days after having received a subcutaneous injection of ~3 nL of pelleted and re-suspended bacteria (**D**) or supernatant (**E**) from the reconstituted BCG-Medac powder. (**F**) Graph showing the proportion of larvae that survived at 5 days post-fertilization, three days after injection with ~3 nL of pelleted, and re-suspended bacteria (blue line) or supernatant (orange line) from the reconstituted BCG-Medac powder. *n* = 15 embryos per group. *: *p* < 0.05.

**Figure 2 cells-12-00508-f002:**
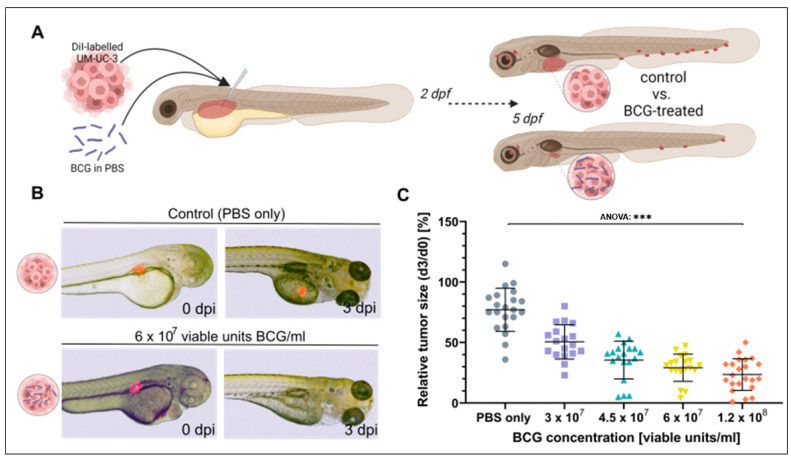
Tumor size regression of UM-UC-3 tumors is observed in zebrafish xenografts after three days of BCG treatment. (**A**) Cartoon illustrating the experimental setup. DiI labelled UM-UC-3 tumor cells mixed with BCG were subcutaneously microinjected into zebrafish larvae 2 days post-fertilization and primary tumor size was monitored until 3 days post injection (dpi). (**B**) Combined light- and fluorescence micrographs of 2-day-old zebrafish larvae implanted with fluorescently labelled primary tumors of control (UM-UC-3 + PBS) and BCG at 0 days post injection (dpi) and 3 dpi. (**C**): Graph showing the relative tumor size of control and BCG-co-injected larvae for 4 different concentrations of BCG-Medac. *n* = 24 injected embryos per group; *** = *p* < 0.0001.

**Figure 3 cells-12-00508-f003:**
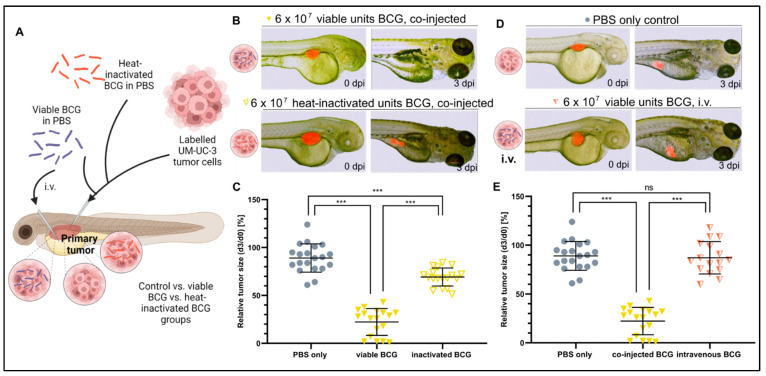
Heat inactivation and intravenous injection of BCG minimize the tumor-killing effect of BCG in UM-UC-3 xenografts. (**A**) Cartoon illustrating the experimental setup. Fluorescently labelled UM-UC-3 tumor cells alone or mixed with either viable (blue) or heat-inactivated (red) BCG were injected into the PVS of 2-day-old zebrafish larvae; alternatively, viable BCG was injected intravenously in subcutaneous tumor-bearing larvae. (**B**) Combined light and fluorescence micrographs at 0 days post injection (dpi) or 3 dpi of fluorescently labelled UM-UC-3 primary tumors mixed with either viable or head-inactivated BCG-Medac and implanted in 2-day-old zebrafish. (**C**) Graph showing the relative tumor size of control and BCG-co-injected larvae from the experiment shown in B. *n* = 24 injected embryos per group; ns = not significant; *** = *p* < 0.0001. (**D**) Combined light and fluorescence micrographs at 0 days post injection (dpi) or 3 dpi of fluorescently labelled UM-UC-3 primary tumors with or without intravenous BCG treatment. (**E**): Graph showing the relative tumor size of control, BCG co-injected, and BCG intravenously injected larvae from the experiment shown in D. *n* = 24 injected embryos per group; ns = not significant; *** = *p* < 0.0001.

**Figure 4 cells-12-00508-f004:**
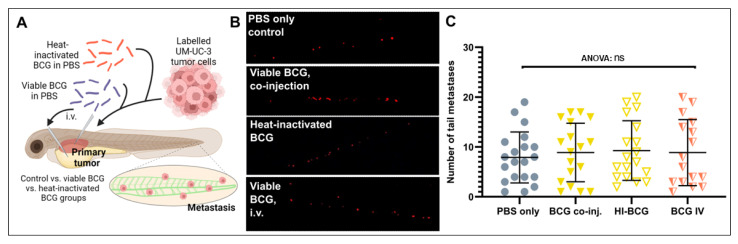
Tumor dissemination is not significantly impacted by BCG treatment of zebrafish UM-UC-3 tumor xenografts. (**A**) Cartoon showing the outline of the experiment. Fluorescently labelled UM-UC-3 tumor cells alone or mixed with viable or heat-inactivated BCG were injected into the PVS of 48 h old zebrafish larvae. Alternatively, viable BCG was injected intravenously in subcutaneous tumor-bearing larvae. (**B**) Fluorescent micrographs of the caudal hematopoietic plexus of control and BCG- (co-)injected tumor-bearing larva at 3 days post-implantation. Red dots represent disseminated tumor cells. (**C**) Graph showing the quantified number of cells present in the caudal hematopoietic plexus in the different treatment groups. *n* = 24 injected embryos per group; ns = not significant.

**Figure 5 cells-12-00508-f005:**
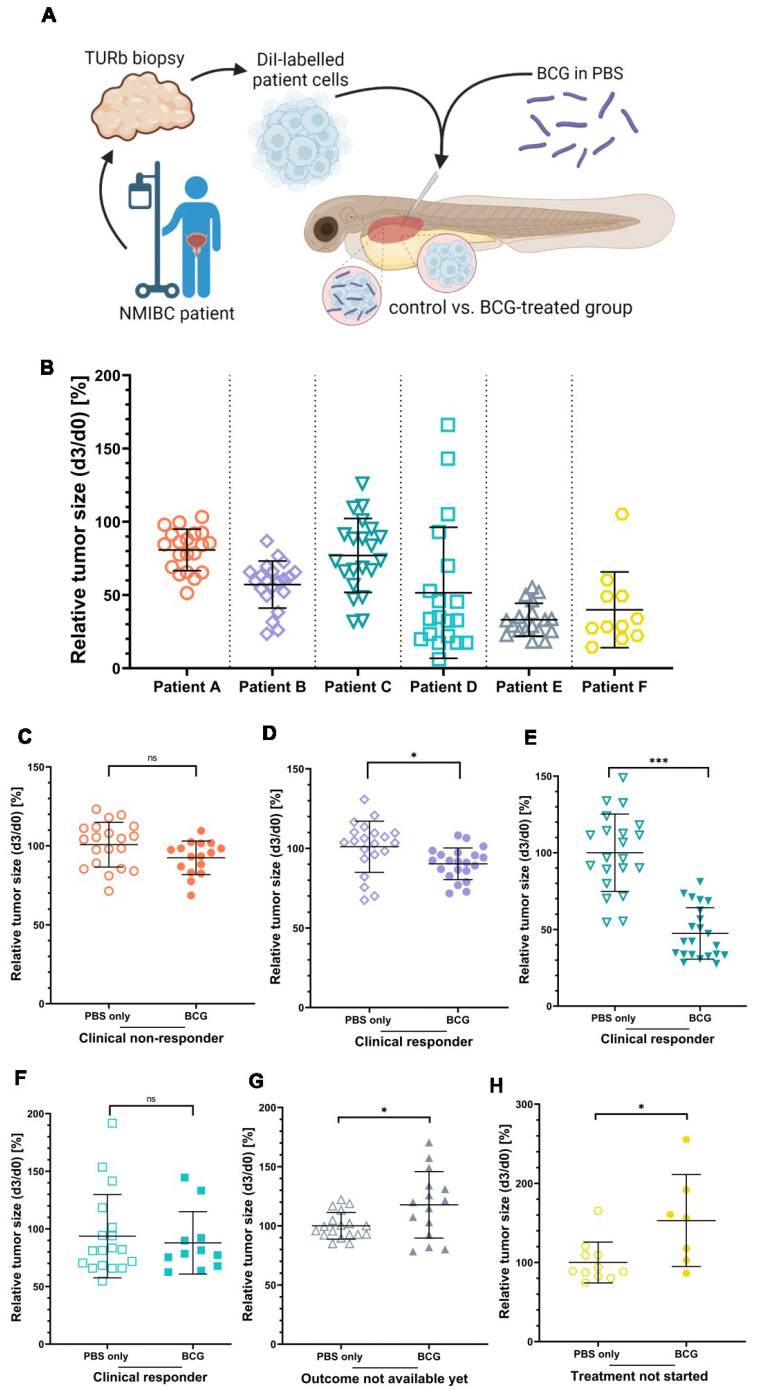
The clinical responses of three NMIBC patients are mirrored by the BCG treatment response of ZTX models. (**A**) Cartoon illustrating the experimental set-up. Cryo-preserved patient tumor tissues were dissociated, fluorescently labelled, and subcutaneously injected into the PVS of 48 h old zebrafish larvae with or without BCG. (**B**) Graph showing the relative tumor size of the six different ZTX models without BCG treatment quantified at 3 days post-injection (dpi). (**C**–**H**) Graphs comparing relative tumor sizes of BCG co-injected and untreated control ZTX groups quantified at 3 dpi. *n* = 20 injected embryos per group; ns = not significant; * = *p* < 0.05; *** = *p* < 0.0001.

**Table 1 cells-12-00508-t001:** Clinical patient data.

Parameter	Value
Number of patients	6
Gender: Male/female	5/1
Age: Median (interval)	76 (66–84)
Tumor stage: T1G2/T1G3/T1G3+CIS	1/3/2

## Data Availability

The data presented in this study are available on request from the corresponding author. The data are not publicly available due to GDPR-regulations.

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
