# Peer review of "Novel Zebrafish Patient-Derived Tumor Xenograft Methodology for Evaluating Efficacy of Immune-Stimulating BCG Therapy in Urinary Bladder Cancer"

_cells, 2023, doi:10.3390/cells12030508_

Round 1

Reviewer 1 Report

Using dissociated patient bladder tumor samples, the authors established the zebrafish tumor xenograft (ZTX) models to identify the efficiency of BCG-therapy. They identified the importance of physical contact between the BCG bacteria and the tumor cells as well as the viability of the bacteria. This ZTX-model might predict BCG response and thereby improve treatment procedure. However, the small number of tumor samples and relatively less clinical data, and lack of rodent animal study reduces the enthusiasm for the study.

Author Response

Comment 1: Using dissociated patient bladder tumor samples, the authors established the zebrafish tumor xenograft (ZTX) models to identify the efficiency of BCG-therapy. They identified the importance of physical contact between the BCG bacteria and the tumor cells as well as the viability of the bacteria. This ZTX-model might predict BCG response and thereby improve treatment procedure. However, the small number of tumor samples and relatively less clinical data, and lack of rodent animal study reduces the enthusiasm for the study.

Response: We appreciate the reviewers supportive and constructive comments. Indeed, we understand the limitations of the study as it pertains to investigations in a non-rodent model and inclusion of only a few patient samples as a first proof-of-principle of the translational relevance. To further strengthen the study, we have included investigations from 3 additional patients (Figure 5) and also a table with patient- and tumor-related data (Table 1), to address the issues of “small number of tumor samples” and “less clinical data” respectively. We believe this significantly strengthen the study and hope the reviewer will agree that this may increase the enthusiasm for the study to some extent.

Reviewer 2 Report

The manuscript covers an interesting topic: the definition of new animal models to predict the effect of intravesical BCG therapy in superficial bladder cancer (BC) patient. As the Authors comment in the manuscript, there are no possibilities to know in advance whether a BC patient will respond or not to the mycobacteria treatment. Thus, new tools are needed to predict refractory BCG-patients. Specifically, the Authors use zebrafish as host for patient-derived tumors. This xenograph animal model consist in injecting directly both tumor cells and BCG. Viable and heat-killed BCG were studied via intratumoral or after intravenously injection. In my opinion the hypothesis is interesting, but the manuscript shows too preliminary results evaluating only one tumor cell line to optimize the model, and three tumor-derived patients. Moreover, several drawbacks in the methodology question the conclusions drawn from the results obtained. In my opinion, there are many factors not considered in the experiments that could be determinant for the results obtained. There is a lack of information related to BCG characteristics, further analysis of the effect/viability of BCG in zebrafish and more robust data are needed to this hypothesis. For further details, please refer to specific comments.

Specific major comments

-  From the results obtained in 1950 to the Morales experiments, there were several assays that corroborated the antitumor effect of BGC. Clarify this point.

-  Were all experiments performed using the same aliquot of BCG? How the concentration between vials of BCG were studied? As the Authors indicate the concentration inside the vial is between 1.88x10^6 to 2.4x10^8. Thus, could be that the differences observed between patients depend on BCG concentration used?

This point is especially relevant when the dose-dependent toxicity effects are explained in the first part of the Results section.

- Did the Authors corroborate the lost of viability of BCG after “heat-inactivation”? BCG could remain viable after 20 min at 80ºC?

- Are BCG growing at 33ºC? whether BCG do not replicate at 33ºC, which is the difference between treating the tumor cells with live BCG at 33ºC and “heat-inactivated BCG”? to discuss properly the results, this point should be clarified.

-  Indicate which is the ratio between tumor cells and cells of BCG injected together.

-  Was the composition of the aliquot of BCG analyzed? BCG is lyophilized in the vials to be preserved. The freezing-dried procedure requires the use of preservative compound that maintain BCG integrity such as trehalose (for instance). Could be these compound the responsible of the “toxicity” showed by BCG? These compounds are water soluble, so it has sense that removing these compounds eliminates the toxicity of “BCG preparation”.

- The appropriate negative control of the experiments should be the content of the vial without BCG.

- The non-toxicity showed by even the higher amount of “cleaned” BCG, could be due to the loss non “activity” of BCG at 33ºC? Discuss this point.

- BCG is able to inhibit UM-UC-3 proliferation in vitro, as the Authors mention in the Discusssion section. The fact that BCG therapy did not impact on tumor cells invasiveness and dissemination could be related to this fact?. Did the Authors tried to inject first the cells and BCG later at the same point of injection?

- Related to this fact, and as the Authors discuss at the end of the first paragraph of the Discussion, tumor cells derived from patients can interact differently to BCG. Do the Authors think that this effect could be observed simply doing 2D in vitro experiments.

- Again, related to this, BCG enters mainly in high-grade tumor cells but BCG do not enter in low-grade tumor cells, in general although there are some exceptions as the Authors explain for UM-UC-3. Do the Authors know which was the grade of tumor cells derived from patients?

- In mammal models (including what happen in clinical setting) the main effect of BCG in vivo is due to the immune system activation. How zebrafish immune system interacts with BCG? Several references describe the immune response in zebrafish after Mycobacterium abscessus or Mycobacterium marinum (both able to growth at 30ºC). Did the Authors know whether BCG is inducing similar responses. In other animal models of bladder cancer, BCG-activated immune system is the key to tackle tumors.

Minor comments:

- Correct “Petri dishes” instead of “petri dishes”

Author Response

Comment: The manuscript covers an interesting topic: the definition of new animal models to predict the effect of intravesical BCG therapy in superficial bladder cancer (BC) patient. As the Authors comment in the manuscript, there are no possibilities to know in advance whether a BC patient will respond or not to the mycobacteria treatment. Thus, new tools are needed to predict refractory BCG-patients. Specifically, the Authors use zebrafish as host for patient-derived tumors. This xenograph animal model consist in injecting directly both tumor cells and BCG. Viable and heat-killed BCG were studied via intratumoral or after intravenously injection. In my opinion the hypothesis is interesting, but the manuscript shows too preliminary results evaluating only one tumor cell line to optimize the model, and three tumor-derived patients. Moreover, several drawbacks in the methodology question the conclusions drawn from the results obtained. In my opinion, there are many factors not considered in the experiments that could be determinant for the results obtained. There is a lack of information related to BCG characteristics, further analysis of the effect/viability of BCG in zebrafish and more robust data are needed to this hypothesis. For further details, please refer to specific comments.

Response: We thank the reviewer for the constructive criticism. We have performed additional studies and revisions of the manuscript to address the specific issues (see details in the list below). As the time to perform revisions was limited (ten days), we have focused on improving the text describing the BCG-related methodology and elaborated on the clinical study as shown in the new Table 1 and revised Figure 5. We hope the reviewer agrees that these revisions and additions have strengthened the manuscript.

Comment 1: From the results obtained in 1950 to the Morales experiments, there were several assays that corroborated the antitumor effect of BGC. Clarify this point.

Response: Further information on the assays investigating the antitumor effect of BCG was added in the manuscript (page 4, lines 10-16).

Comment 2: Were all experiments performed using the same aliquot of BCG? How the concentration between vials of BCG were studied? As the Authors indicate the concentration inside the vial is between 1.88x10^6 to 2.4x10^8. Thus, could be that the differences observed between patients depend on BCG concentration used? This point is especially relevant when the dose-dependent toxicity effects are explained in the first part of the Results section.

Response: The same lyophilized vial of BCG was used for all experiments, meaning that the concentration should be the same for the dose-dependent toxicity, efficacy and patient experiments. The concentration was calculated based on the mean concentration within the concentration-rage of the vial, which has now been clarified in the materials and methods section (page 8 line 7-2 from the bottom of the page).

Comment 3: Did the Authors corroborate the lost of viability of BCG after “heat-inactivation”? BCG could remain viable after 20 min at 80ºC?

Response: We agree with the reviewer that a small fraction of the BCG potentially could potentially have remained viable after heat-inactivation for 20 min at 80ºC. We did not test this experimentally ourselves as wet used a previously validated protocol where this had been thoroughly tested. The protocol was adopted from Sabiiti et al (sabiiti et al, j. clin. Microbiol, 2019,. 57(4), pp. e01778-18), who found that heat-inactivation of M. tuberculosis bacilli as well as M. bovis BCG cultures by boiling at 20 minutes at 80°C efficiently led to a completely lack of bacterial growth during 42 days of culture, compared to non-treated cultures that expanded to 21x the initial optical density during this culture period. This clearly showed that heat inactivation at 80 ºC for 20 min. led to a loss of bacterial viability. We have now in the revised version of the manuscript clarified a) the reason for choosing the specific heat-inactivation protocol and b) the fact that we can only presume that the majority of the BCG bacteria were inactivated in the process (page 9, line 14-16).

Comment 4: Are BCG growing at 33ºC? whether BCG do not replicate at 33ºC, which is the difference between treating the tumor cells with live BCG at 33ºC and “heat-inactivated BCG”? to discuss properly the results, this point should be clarified.

Response: The reviewer brings up an important point. Indeed, BCG bacteria have been characterized to retain its capacity to be internalized and thereby to activate cellular immune responses at 33°C (Acosta et al, PLoS Pathogenesis, 2018, 14(7), e1007151), a temperature that is 2,5°C lower than the temperature used in our study (35,5°C). The bacteria will also retain viability and replicative capabilities under such temperatures (Barbier et al, PLoS ONE, 2017, 12(4), e0176315). These points have now been made in the materials and methods section of the revised manuscripts (page 9 line 5-7). On the issue of the mechanism of action and why the bacteria have to be viable to have anti-tumor activity, this is a question of ongoing research and currently poorly understood. We have started to investigate the immune-activating mechanisms elicited by BCG and other immune-oncological drugs in more detail, but those results are planned to be published separately as they are too complicated to be appropriately explained within the current manuscript (see also our reply to comment 12 below).

Comment 5: Indicate which is the ratio between tumor cells and cells of BCG injected together.

Response: We agree with the reviewer that this is important information to include in the manuscript. The ratios have been calculated and included as Supplemental Table 1, and also inserted below.

Supplemental Table 1: Calculated BCG:cell ratios for each concentration:

BCG concentration / ml

Cell number/nl

BCG:cell ratio

3 x 107

~300 cells

~1:10

4.5 x 107

~1:7.5

6 x 107

~1:5

1.2 x 108

~1:2.5

Comment 6: Was the composition of the aliquot of BCG analyzed? BCG is lyophilized in the vials to be preserved. The freezing-dried procedure requires the use of preservative compound that maintain BCG integrity such as trehalose (for instance). Could be these compound the responsible of the “toxicity” showed by BCG? These compounds are water soluble, so it has sense that removing these compounds eliminates the toxicity of “BCG preparation”.

Response:  We thank the reviewer for raising this possibility. According to the supplier of the BCG used in this study, the ingredients contained within the lyophilized powder, other than the BCG bacteria themselves, are: polygeline, glucose anhydrous and polysorbate 80. Even though trehalose is the most commonly used stabilizer, glucose was used as the protective agent of the bacilli in this formulation. This information has been clarified in the materials and methods section on page 8 line 6 from the bottom of the page. Glucose and polygeline are not causing harmful effects when injected in zebrafish larvae, and therefore unlikely to be responsible for the observed toxicity.

Comment 7: The appropriate negative control of the experiments should be the content of the vial without BCG.

Response: We agree with the reviewer and this is also what has been done. As the water-soluble compounds were “washed out” due to their toxicity, the remaining content of the vial was the bacteria resuspended in PBS, and indeed we used PBS as the neg

Comment 8: The non-toxicity showed by even the higher amount of “cleaned” BCG, could be due to the loss non “activity” of BCG at 33ºC? Discuss this point.

Response: We see the reviewers point. Both “cleaned” and non-cleaned BCG were incubated at 35.5°C in the zebrafish and our view is that, if the BCG has lost its activity at this temperature, it should also not have shown toxicity in the non-cleaned group. Furthermore the cleaned BCG, if it had lost its activity, should not have had an impact on the tumor growth as seen in the subsequent experiments. These arguments have now been added to the discussion on page 17 line 5-2 from the bottom of the page.

Comment 9: BCG is able to inhibit UM-UC-3 proliferation in vitro, as the Authors mention in the Discusssion section. The fact that BCG therapy did not impact on tumor cells invasiveness and dissemination could be related to this fact?. Did the Authors tried to inject first the cells and BCG later at the same point of injection?

Response: We thank the reviewer for bringing up the important question of the inter-relationship between cell proliferation and metastasis. Indeed, we have also seen using other types of drugs on bladder cancer samples (MVAC and GC) that cytostatic or cytotoxic efficacy is poorly correlated with anti-metastatic effects. Inhibition of proliferation does not automatically lead to maintained invasiveness, rather the two processes seems to be un-associated, at least in the ZTX models. In initial studies we also tried injecting BCG i.v. 24 hours after tumor implantation, but as we did not see any effect of BCG injected i.v. immediately after tumor implantation (Figure 3 and Figure 4), the 24h delay did not change that. We reasoned that this was likely due to the i.v. administration which did not allow physical contact between the BCG bacteria and the tumor cells, and did therefore not include that data in the manuscript. It has been established in the literature (and discussed above as well as in the manuscript) that internalization of the BCG bacteria are required for efficacy. It is theoretically possible that the metastasized cells escaped the primary tumor environment quickly after implantation, prior to internalizing the bacteria. It is also possible that mechanisms that in some tumor cells lead to inhibition of bacterial internalization may also be involved in metastasis, alternatively that immune-evasion mechanisms allowing tumor cells with internalized bacteria to stay alive could also lead to increased invasiveness. In all these cases, injecting the bacteria later would not help to answer this question, but this will be an interesting point to investigate in a dedicated follow-up studies in the future. This hypothesis has not been included in the discussion on page 18, line 2 from the bottom of the page - page 19 line 17.

Comment 10: Related to this fact, and as the Authors discuss at the end of the first paragraph of the Discussion, tumor cells derived from patients can interact differently to BCG. Do the Authors think that this effect could be observed simply doing 2D in vitro experiments.

Response: The reviewer brings up an important point. Indeed, direct drug-induced cytotoxic effects which do not depend on other aspects of the tumor microenvironment would be clear also from simpler 2D in vitro experiments. However, mechanisms wherein the tumor microenvironment, including anti-tumor immunity, plays an important role, are not fully recapitulated within such simplified systems. As we discuss in the introduction, 3D models that include host immunity, such as the ZTX system used in this manuscript, are required to study tumor-immune interactions involved in (for example) anti-tumor efficacy of BCG. The paragraph in the discussion which was mentioned by the reviewer has now been amended to include this argument (page 16 line 4-7).

Comment 11: Again, related to this, BCG enters mainly in high-grade tumor cells but BCG do not enter in low-grade tumor cells, in general although there are some exceptions as the Authors explain for UM-UC-3. Do the Authors know which was the grade of tumor cells derived from patients?

Response: We appreciate the reviewers comment. We have now added data on all patients and their disease to Table 1 of the revised version of the manuscript. As the reviewer will see, five of the tumors used were T1G3 (grade 3) and one was T1G2 (grade 2). The majority of the tumors were therefore indeed high grade.

Comment 12: In mammal models (including what happen in clinical setting) the main effect of BCG in vivo is due to the immune system activation. How zebrafish immune system interacts with BCG? Several references describe the immune response in zebrafish after Mycobacterium abscessus or Mycobacterium marinum (both able to growth at 30ºC). Did the Authors know whether BCG is inducing similar responses. In other animal models of bladder cancer, BCG-activated immune system is the key to tackle tumors.

Response: The reviewer points to a very important and woefully under-studied question related to BCG therapy. We have started investigating the role of macrophages and neutrophils specifically in the context of the anti-tumor response to BCG in the ZTX models, but the data was so-far inconclusive and require further experimentation to allow an in-depth understanding. We know, however, that adaptive immunity likely does not play a role in the responses observed in the ZTX models (although they may play an additional role, augmenting the efficacy in the patients), as the larvae do not have B- or T-cells, and as the tumor cells will not have been exposed to BCG prior to implantation. Initial studies demonstrated that zebrafish macrophages and neutrophils were recruited to the tumor area following implantation, but in our preliminary studies we found that this recruitment was similar in non-treated as in BCG-treated tumors. We are now investigating the activation profile of the macrophages and neutrophils, performing studies in knock-out strains devoid of these cell types, and plan gain-of-function studies using patient-derived macrophages or neutrophils added to the xenografts instead of BCG. We hope to be able to publish these studies during 2023. At present we are, however, not able to elaborate further on the mechanisms involved in the BCG efficacy in the ZTX models within the current manuscript as that would be too preliminary and jeopardize our future publication of the data.

Comment 13: Correct “Petri dishes” instead of “petri dishes”

Response: The typos have now been corrected.

Reviewer 3 Report

The manuscript by Saskia Kowald et al described a method to evaluate the efficacy of immune-stimulating BCG therapy in urinary bladder cancer using zebrafish. They found that BCG-treatment by co-injection with the tumor cells could result in significant and dose-dependent primary tumor size regression. Their results indicates that ZTX-models might predict BCG response and thereby improve treatment planning. This manuscript is well organized and suitable for publication on Cells after a minor revision. 

Minor points:

1.     We know that the zebrafish embryos will cause deformities if the embryos were incubated for a long time at high temperature, here the temperature the paper used is 35.5 ℃, so do you see any malformation?

2.     For some figures, the authors should label the significance clearly, for example, Fig2c, how about PBS only and 3 X 107, if no significant, you should label “ns”.

Author Response

Comment: The manuscript by Saskia Kowald et al described a method to evaluate the efficacy of immune-stimulating BCG therapy in urinary bladder cancer using zebrafish. They found that BCG-treatment by co-injection with the tumor cells could result in significant and dose-dependent primary tumor size regression. Their results indicates that ZTX-models might predict BCG response and thereby improve treatment planning. This manuscript is well organized and suitable for publication on Cells after a minor revision. 

Response: We thank the reviewer for the highly uplifting and supportive comments and succinct summary of our findings.

Comment 1:  We know that the zebrafish embryos will cause deformities if the embryos were incubated for a long time at high temperature, here the temperature the paper used is 35.5 â„ƒ, so do you see any malformation?

Response: The reviewer raises an important question. Indeed temperatures of 35.5 degrees may cause transient pericardial edema when embryos are treated with such temperatures from the 1-cell stage onwards. When treatment starts at two days post fertilization and lasts for three days, we do, however, not see any noticeable malformations or other toxicity phenotypes due solely to the temperature. This has in the revised version of the manuscript been clarified and stated explicitly on page 17 line 14-18. We thank the reviewer for noticing this lack of clarity.

Comment 2. For some figures, the authors should label the significance clearly, for example, Fig2c, how about PBS only and 3 X 107, if no significant, you should label “ns”.

Response: We fully agree with the reviewer and apologize for the ambiguity in noting the absence of significance in some of the figures. This has now been corrected in the revised version of the manuscript (figure 2 and 4).

Round 2

Reviewer 2 Report

The reviewer stills thinks that the topic is relevant, and the approach is interesting, but the manuscript has still many incongruences related to the applicability of this model to analyses the effect of BCG on bladder cancer treatment.

Whether the immune system is not considered, the usefulness of this model for the study of BCG as antitumor agent is not clear. The Authors are studying a mycobacteria in which its main role is to trigger a proper immune response to fight tumors.

Whether the critical point, as the Authors observed, is to put together tumor cells and BCG, only in vitro studies are enough to elucidate the effect that the Authors observe in the zebrafish (as the Authors remark in the answers to the reviewer comments). In my opinion, the model needs to be completed with the analysis of the immune response. If the zebrafish immune system (innate response) or patient-derived cells (as indicated in the response to comment 12) has a role in reducing tumor growth after BCG treatment, the study merits then a high recognition. But this plausible role of the immune system needs to be demonstrated in this model, before being accepted.

From the responses of the Authors new questions/comments have raised. That is:

-        Response to Comment 1: The reference added in the introduction section (reference 14) is not related to BCG treatment. The reference 14 is related to BGG: bovine y-globulin (BGG, Armour Pharmaceutical Co., Kankakee, Illinois)!!

-        Response to Comment 2: if the Authors use only one BCG vial for all experiments, how the Authors control the stability of the diluted vial??? Was the BCG frozen? How? It was a lost of viability when it was frozen??

-        Response to Comment 4: The reference Barbier et al, PLoS ONE, 2017, 12(4), e0176315, do not show that bacteria grow at 33ºC. This reference show that mycobacteria remain viable after incubating several weeks at 22ºC, but the bacteria do not grow at such temperatures.

-        Response to Comment 6: how the Authors know that polygeline, glucose anhydrous, and polysorbate 80 are not toxic for zebrafish? This should be the right control for the study of the toxic compound of the BCG vial. The correct control to test the effect of the compound responsible of toxicity is the exact content of the commercial vial without the bacteria.

-        Response to Comment 10. For this reason, the Reviewer believe that it is crucial to analyse the possible role of the innate immune system of zebrafish in these experiments. If not, in vitro experiments could be enough to see the effect of BCG in patient-derived cells.

-        Response to Comment 11. NMIBC patients should not be included whether the outcome of the BCG instillations are not resolved. It is not possible to interpret the results.

Author Response

Initial comment: The reviewer stills thinks that the topic is relevant, and the approach is interesting, but the manuscript has still many incongruences related to the applicability of this model to analyses the effect of BCG on bladder cancer treatment.

Whether the immune system is not considered, the usefulness of this model for the study of BCG as antitumor agent is not clear. The Authors are studying a mycobacteria in which its main role is to trigger a proper immune response to fight tumors.

Whether the critical point, as the Authors observed, is to put together tumor cells and BCG, only in vitro studies are enough to elucidate the effect that the Authors observe in the zebrafish (as the Authors remark in the answers to the reviewer comments). In my opinion, the model needs to be completed with the analysis of the immune response. If the zebrafish immune system (innate response) or patient-derived cells (as indicated in the response to comment 12) has a role in reducing tumor growth after BCG treatment, the study merits then a high recognition. But this plausible role of the immune system needs to be demonstrated in this model, before being accepted.

Response: We appreciate the reviewers interest in understanding the mechanisms underlying the immune activation observed in BCG treated zebrafish. We have done some experiments in this area, but realized that the mechanisms were less straight forward than what we initially hoped. For example, we could not find increased infiltration of macrophages in the tumors in BCG-treated lyzEGFP (macrophage reporter strain) zebrafish larvae compared to control larvae (see image in the attachment). This suggested that activation or polarization of the macrophages may be involved rather than simply recruitment. We are currently looking into this in more detail. We will also do loss-of-function experiments by knocking out genes important for macrophage development (spi1) leading to macrophage-deficient zebrafish, to see if they now loose the ability to respond to BCG, and we will co-implant tumor cells with human macrophages stimulated prior to implantation to become either classically (M1) or alternatively (M2) activated, to see if these subsets could increase the BCG effect. Finally we will molecularly characterize macrophages from responding and non-responding patients to identify which macrophage subtypes are important for the response. Similar studies will be done for Neutrophils as well. The results from these experiments will likely inspire further experiments needed to gain full insights into the important question of how the immune system is involved in mediating BCG efficacy. These studies are in our opinion extending beyond the scope of this manuscript. First of all, the current manuscript was submitted to a special issue on new methods for 3D tumor cultures, and an extensive investigation on immune-mechanisms is out of scope for this issue. Second, the manuscript would be too large if all the coming immune-data was to be included, and it would loose coherence as it would deal with several questions in one and the same article. Thirdly, those experiments will take time to complete and that is not possible within the short review period required for the special issue. We therefore sincerely hope the reviewer will see these issues and agree that the complex but highly interesting mechanisms underlying activation of the immune system related to BCG efficacy deserves proper and careful evaluation in a manuscript of its own in the (near) future.

Whether the zebrafish xenograft system has benefits beyond in vitro cell culture studies when it comes to evaluating BCG efficacy, is discussed in several parts of the manuscript. Firstly, establishing cell cultures from TUR-B biopsies takes a long time, as primary tumor cells grow slowly. This will also fail for approximately half of the patients, whose tumors will never grow. Zebrafish xenograft models were created, BCG efficacy evaluated and the results analyzed within 5 days and worked for all patients included in the study. Zebrafish tumor models also allow evaluation of metastasis – as shown in our manuscript – which the 3D culture systems do not. Finally, should our thorough investigations into the immune systems role in the BCG response show that patient-immune cells found in the tumors are important, these are also part of the tumor microenvironment in the zebrafish xenogaft model but will die or be competed out during the cell culture process and therefore not part of in vitro systems. These benefits are likely going to be critical if a functional companion diagnostic test for testing BCG efficacy are going to be clinically implemented which would therefore be possible with the zebrafish system, but not in vitro cell culture systems.

Comment 1: The reference added in the introduction section (reference 14) is not related to BCG treatment. The reference 14 is related to BGG: bovine y-globulin (BGG, Armour Pharmaceutical Co., Kankakee, Illinois)!!

Response: We deeply apologize for the mistake and thank the reviewer for his/her careful scrutiny of our manuscript which allowed this mistake to be caught in time. We have now exchanged “BCG” for “immunogenic antigens” to reflect what was studies in the paper (page 4, line 12).

Comment 2: if the Authors use only one BCG vial for all experiments, how the Authors control the stability of the diluted vial??? Was the BCG frozen? How? It was a lost of viability when it was frozen??

Response: The BCG vial contained lyophilized powder that was stored according to the manufacturers instructions such that the stability and activity of the bacteria were maintained throughout the duration of the study. We weighed the necessary amount of powder and reconstituted it as described in the material and methods prior to each experiment. The ampule was re-sealed after each experiment. As such, bacteria were not frozen or in other ways handled in a manner that could impair their activity or survival. This has now been clarified in the materials and methods section, page 8 line 4 from the bottom of the page – page 9 line 1.

Comment 3: The reference Barbier et al, PLoS ONE, 2017, 12(4), e0176315, do not show that bacteria grow at 33ºC. This reference show that mycobacteria remain viable after incubating several weeks at 22ºC, but the bacteria do not grow at such temperatures.

Response: We agree with the reviewer that Barbier et al did not show growth of the bacteria but rather that they remain viable. This is also what is relevant for the BCG therapy as the bacteria do not need to proliferate to have an effect – they only need to be viable. Indeed, this is the foundation for why the bacteria could be attenuated (so the only grow very slowly) without this having consequences for the efficacy of the treatment. We have now removed “can replicate” from the sentence such that the reference only is used to support that the bacteria remains viable (page 9 line 8).

Comment 4: how the Authors know that polygeline, glucose anhydrous, and polysorbate 80 are not toxic for zebrafish? This should be the right control for the study of the toxic compound of the BCG vial. The correct control to test the effect of the compound responsible of toxicity is the exact content of the commercial vial without the bacteria.

Response: Polygeline is rapidly metabolized and therefore unlikely to be the agent causing toxicity. Furthermore it consists of urea and gelatine-like polypeptides, both of which are not toxic in zebrafish. Glucose has been used frequently in zebrafish studies and is not causing toxicity in the concentrations used here. Polysorbate 80 is, however, toxic in zebrafish. This is described in the discussion on page 7 line 8-7 from the bottom of the page. As the lyophilized BCG powder only contain these molecules and the bacteria, and as all three molecules are removed with the supernatant as the bacteria are “cleaned” prior to use, what remains is bacteria and PBS. Indeed, the control group was given PBS instead of the bacteria+PBS used in the BCG-group. The commercial vial could not be used as a control as that was toxic due to the presence of polysorbate 80, as shown in figure 1.

Comment 5: For this reason, the Reviewer believe that it is crucial to analyse the possible role of the innate immune system of zebrafish in these experiments. If not, in vitro experiments could be enough to see the effect of BCG in patient-derived cells.

Response: See our comment to your initial statement.

Comment 6: NMIBC patients should not be included whether the outcome of the BCG instillations are not resolved. It is not possible to interpret the results.

Response: We agree with the reviewer and made the same assessment when we prepared the manuscript, which is why only three patients were included. Adding more patients was, however, requested by several reviewers, who have now – after this data was added – approved the manuscript. We are therefore not able to remove those extra patients at this stage, as that would be unethical and dishonest towards the other reviewers as their approval was contingent on the addition of this data. We sincerely hope the reviewer can see our conundrum and accept that the new patient data stays in the manuscript.
